# Effect of GLP-1 Receptor Agonist on Ischemia Reperfusion Injury in Rats with Metabolic Syndrome

**DOI:** 10.3390/ph17040525

**Published:** 2024-04-19

**Authors:** Marko Ravic, Ivan Srejovic, Jovana Novakovic, Marijana Andjic, Jasmina Sretenovic, Maja Muric, Marina Nikolic, Sergey Bolevich, Kirill Alekseevich Kasabov, Vladimir Petrovich Fisenko, Aleksandra Stojanovic, Vladimir Jakovljevic

**Affiliations:** 1Department of Pharmacy, Faculty of Medical Sciences, University of Kragujevac, Svetozara Markovica 69, 34000 Kragujevac, Serbia; markoravic@hotmail.com (M.R.); jovana.ilona@gmail.com (J.N.); andjicmarijana10@gmail.com (M.A.); 2Center of Excellence for the Study of Redox Balance in Cardiovascular and Metabolic Disorders, University of Kragujevac, Svetozara Markovica 69, 34000 Kragujevac, Serbia; ivan_srejovic@hotmail.com (I.S.); drj.sretenovic@gmail.com (J.S.); majanikolickg90@gmail.com (M.M.); marina.rankovic.95@gmail.com (M.N.); drvladakgbg@yahoo.com (V.J.); 3Department of Physiology, Faculty of Medical Sciences, University of Kragujevac, Svetozara Markovica 69, 34000 Kragujevac, Serbia; 4Department of Pharmacology, First Moscow State Medical University I.M. Sechenov, Trubetskaya Street 8, Str. 2, 119991 Moscow, Russia; kasabov_k_a@staff.sechenov.ru (K.A.K.); fisenko_v_p@staff.sechenov.ru (V.P.F.); 5Department of Human Pathology, First Moscow State Medical University I.M. Sechenov, Trubetskaya Street 8, Str. 2, 119991 Moscow, Russia; bolevich2011@yandex.ru

**Keywords:** metabolic syndrome, GLP-1 agonists, oxidative stress, cardioprotection

## Abstract

Metabolic syndrome (MetS) represents an important factor that increases the risk of myocardial infarction, and more severe complications. Glucagon Like Peptide-1 Receptor Agonists (GLP-1RAs) exhibit cardioprotective potential, but their efficacy in MetS-related myocardial dysfunction has not been fully explored. Therefore, we aimed to assess the effects of exenatide and dulaglutide on heart function and redox balance in MetS-induced rats. Twenty-four Wistar albino rats with induced MetS were divided into three groups: MetS, exenatide-treated (5 µg/kg), dulaglutide-treated (0.6 mg/kg). After 6 weeks of treatment, in vivo heart function was assessed via echocardiography, while ex vivo function was evaluated using a Langendorff apparatus to simulate ischemia-reperfusion injury. Heart tissue samples were analyzed histologically, and oxidative stress biomarkers were measured spectrophotometrically from the coronary venous effluent. Both exenatide and dulaglutide significantly improved the ejection fraction by 3% and 7%, respectively, compared to the MetS group. Histological analyses corroborated these findings, revealing a reduction in the cross-sectional area of cardiomyocytes by 11% in the exenatide and 18% in the dulaglutide group, indicating reduced myocardial damage in GLP-1RA-treated rats. Our findings suggest strong cardioprotective potential of GLP-1RAs in MetS, with dulaglutide showing a slight advantage. Thus, both exenatide and dulaglutide are potentially promising targets for cardioprotection and reducing mortality in MetS patients.

## 1. Introduction

Despite great scientific interest and intensive research into the phenomenon of myocardial ischemia and reperfusion, there is a large number of unknowns that make it impossible to understand the molecular mechanisms and signaling cascades that mediate heart tissue damage due to reduced blood flow. Due to the well-known fact that cardiovascular diseases (CVD) are the leading cause of morbidity and mortality worldwide [1], a large amount of research is focused on finding ways to reduce myocardial damage. Myocardial infarction (MI), as one of the most common cardiovascular disorders, but also the most deleterious, occurs due to a limited or blocked myocardial blood supply. Metabolic syndrome (MetS) and diabetes represent important factors that increase the risk of MI, as well as the probability of greater myocardial damage and more severe complications [2]. MetS can encompass visceral obesity, insulin resistance, glucose intolerance, dyslipidemia, and hypertension.

Disruption of the blood flow results in ischemia, reduced oxygen and nutrient inflow; it induces tissue necrosis, inflammation, oxidative damage, and disturbances in heart function [3,4]. Contrary to expectation, the reestablishment of blood flow through the coronary artery causes additional damage to the ischemic myocardium, leading to the phenomenon of reperfusion injury [5]. Ischemic damage of the myocardium and following reperfusion injury combined form phenomena known as ischemia/reperfusion injury (IRI). Various changes in heart physiology due to IRI create a complex network of cascades and pathways which interact with each other at different stages, including disturbance in membrane ion transport, changes in intracellular calcium content, oxidative stress due to the increased production of reactive oxygen species (ROS), augmented inflammatory response, and changes of mitochondrial activity [3,4,6,7,8]. Therefore, there are a number of potential therapeutic targets to minimize the myocardial damage induced by IRI.

A great effort is constantly invested in finding the best possible protective mechanism that would prevent short-term and long-term heart function disorders after IRI. One of the first mechanisms of myocardial protection against IRI was ischemic conditioning, which implies multiple short-term periods of ischemia preceding long-term ischemia [9]. Given the recent knowledge generated about the role of certain mediators and signaling pathways in IRI, a large amount of research dealt with the effectiveness of certain pharmacological agents in minimizing myocardial damage [10,11,12,13].

Oxidative stress, usually defined as a discrepancy in the production and elimination of reactive species (mostly oxygen, but also nitrogen, sulfur or carbon), in favor of their accumulation, has a key role in many CVD [14]. Oxidative stress in diabetes and MetS is additionally augmented, which is why it is one of the key factors for the development of cardiovascular complications associated with impaired glucoregulation [15]. Oxidative stress is in close relation to inflammation. It was shown that GLP-1RAs can modulate the inflammatory response through the inhibition of pro-inflammatory cytokine production such as such as interleukin-6 (IL-6) and tumor necrosis factor alpha (TNF-α) [16]. Another mechanism, according to previous studies, is a reduction in leukocyte adhesion and infiltration into the myocardial tissue. These anti-inflammatory actions can diminish the severity of inflammation-induced myocardial damage during reperfusion [17].

The glucagon-like peptide-1 (GLP-1) is a hormone that plays a key role in glucose homeostasis, reducing insulin secretion and increasing insulin sensitivity, as well as regulating food intake, and improving lipid metabolism [18]. GLP-1 receptor agonists (GLP-1RAs) are a class of drugs that mimic the effects of GLP-1 and in addition to their glucose-lowering effects GLP-1RAs have been shown to have cardioprotective effects [18]. It was recently shown that liraglutide, as a GLP-1RA, improves mitochondrial function and reduces oxidative stress in rat hearts and cardiomyocytes [19]. On the other hand, another study showed that the application of GLP-1RAs could improve heart function and normalize cardiac enzyme levels in the experimental model of IRI [20].

Given that IRI is the core phenomenon in pathogenesis of MI, where MI is a key health problem of modern medicine due to its high prevalence and associated disorders, there is a great need for a deeper understanding of IRI and the development of new strategies for reducing its consequences. Furthermore, due to the fact that MetS significantly increases the risk of MI and other CVD, and that the prevalence of MetS is increasing, the objective of this research was the changes induced by IRI in hearts of rats with MetS, and the ability of GLP-1RAs in their attenuation. The aim of this study was to assess and compare the effects of two GLP-1RAs, exenatide and dulaglutide, on the heart function and redox balance in the experimental model of IRI in rats with MetS using the Langendorff technique of an isolated mammalian heart.

## 2. Results

### 2.1. Effects of GLP-1RAs on Weight, Glycemia, Insulin Levels, and Systolic Blood Pressure

Body weight was measured at the beginning, in the 3rd week, and at the end of the experiment. In all groups, there was a statistically significant increase between points of interest but the increase in body weight in the GLP-1RA-treated groups was less pronounced (weight (g) at the beginning, in the third week and at the end of the experiment, respectively—MetS group—303.57, 374.29, 394.29; Exenatide—309.29, 350, 360.71; Dulaglutide—301.67, 346.67, 350.83) (Figure 1A).

As we expected, both applied drugs exenatide and dulaglutide significantly reduced blood glucose levels after 3 weeks of treatment, as well as at the end of the experimental protocol, while in the MetS group we did not notice any changes (glycemia (mmol/L) before drug application, after 3 weeks of treatment, and at the end of the experiment, respectively—MetS—16.52, 16.22, 17.2; Exenatide—16.25, 9.09, 7.75; Dulaglutide—15.2; 9.68, 8.1) (Figure 1B).

Values of glycemia during an oral glucose tolerance test were significantly lower in both groups treated with GLP-1RAs compared to the MetS group, in all points of measurement (0, 30, 60, 90, 120, 180 min). However, there were no significant differences in glucose levels between rats receiving exenatide and dulaglutide (glycemia (mmol/L) during OGTT measured at 0, 30, 60, 90, 120, 180 min, respectively—MetS—17.20, 22.18, 23.17, 21.60, 20.08, 18.87; Exenatide—7.75, 12.87, 12.20, 10.8, 9.28, 8.38; Dulaglutide—8.1, 13.07, 12.55, 11.65, 10.55, 9.35) (Figure 1C).

Insulin concentration was also measured during the OGTT. In all three groups, the fasting insulin levels (μIU/mL) were significantly lower compared to the values 3 h after glucose administration (measured at 0 and 180 min), respectively—MetS—7.83, 10.32; Exenatide—4.23, 6.49; Dulaglutide—3.99, 5.47) (Figure 1D).

We measured systolic blood pressure as one of the parameters for the confirmation of MetS induction and also after the chronic administration of drugs. Only dulaglutide reduced systolic blood pressure (SBP) significantly. SBP was reduced in the exenatide group also, but this reduction was not statistically significant (SBP (mmHg) before the drug application and at the end of the experiment, respectively—MetS—170.33, 174.67; Exenatide—178.5, 165.67; Dulaglutide—173.33, 146.5) (Figure 1E).

### 2.2. Effects of GLP-1RA on In Vivo Cardiac Function

In vivo monitoring of cardiodynamic changes revealed that both groups with applied GLP-1RAs improved their echocardiographic parameters, i.e., interventricular septal wall thickness at end-systole and end-diastole (IVSs and IVSd), left-ventricular posterior wall thickness at end-systole and end-diastole (LVPWs and LVPWd), and left-ventricular internal diameter at end-systole (LVIDs) were decreased by exenatide and dulaglutide compared to MetS, while LVPWd produced IVSs reduction in GLP-1RA-treated groups in favor of dulaglutide. An increase in the ejection fraction was observed in the dulaglutide group compared to the MetS and exenatide group (Table 1).

### 2.3. Effects of GLP-1RAs on Cardiodynamic Parameters

Parameters of cardiac contractility presented as the maximum and minimum rates of pressure development in the left ventricle (dp/dt max and dp/dt min) are significantly increased in groups treated with GLP-1RAs at the end of the stabilization period as well as at the last minute of reperfusion compared to the MetS group (dp/dt max values (mmHg/s) at the end of stabilization and in the last minute of reperfusion, respectively—MetS—1690.25, 1517.98; Exenatide—2138.6, 2297.66; Dulaglutide—2230.26, 2400.8; dp/dt min values (mmHg/s) at the end of stabilization and in the last minute of reperfusion, respectively—MetS—−1135.05, −902.57; Exenatide—−1596.97, −1611.46; Dulaglutide—−1563.82, −1611.36) (Figure 2A,B). Also, in the MetS group, a markedly lower value of dp/dt max was observed in 60 min of reperfusion compared to the value before ischemia, while the opposite results were noticed in both of the treated groups (Figure 3A). Similar results are seen for systolic left-ventricular pressure (SLVP); in the MetS group significantly lower values were seen at the last minute of reperfusion compared to the stabilization period. A more pronounced increase in SLVP values was seen between the treated and untreated group, in favor of the exenatide and dulaglutide group (*p* < 0.001) (SLVP values (mmHg) at the end of stabilization and in the last minute of reperfusion, respectively—MetS—39.45, 32.22; Exenatide—60.5, 58.51; Dulaglutide—64.3, 59.62) (Figure 2C). Noticeable changes in DLVP only differ at the stabilization period in the dulaglutide group compared to MetS (DLVP values (mmHg) at the end of stabilization and in the last minute of reperfusion, respectively—MetS—5.70, 5.58; Exenatide—5.41, 5.31; Dulaglutide—5.1, 5.48) (Figure 2D). Recovery of heart rate (HR) with statistical difference was observed in both treated groups compared to MetS at the last minute of reperfusion (HR (bpm), at the end of stabilization, and in the last minute of reperfusion, respectively—MetS—259, 202; Exenatide—262.43, 256; Dulaglutide—264.4, 259.4) (Figure 2E and Figure 3E). Coronary flow increased in the last minute of reperfusion compared to the stabilization period in the dulaglutide group, while in the other two groups there was not a statistically significant difference (CF values (mL/min) at the end of stabilization and in the last minute of reperfusion, respectively—MetS—12.6, 13.2; Exenatide—12.66, 13.34; Dulaglutide—11.64, 13.88) (Figure 3F).

### 2.4. Effects of GLP-1RAs on Oxidative Stress

Values of index of lipid peroxidation measured as TBARS did not change significantly among all three groups (TBARS values (µmol/mL) at the end of stabilization and in the last minute of reperfusion, respectively—MetS—14.18, 17.23; Exenatide—14.09, 14.21; Dulaglutide—14.34, 15.45) (Figure 4A and Figure 5A). The level of nitrites (NO_2_^−^) was considerably increased in both investigated GLP-1RAs, exenatide, and dulaglutide in the last minute of the reperfusion period and in the dulaglutide group at the stabilization point compared to MetS (NO_2_^−^ values (nmol/mL) at the end of stabilization and in the last minute of reperfusion, respectively—MetS—131.36, 132.89; Exenatide—164.3, 175.85; Dulaglutide—183.93, 203.76) (Figure 4B). In coronary venous, effluent super oxide anion radical (O_2_^−^) was significantly decreased at the end of the stabilization period and then compared the exenatide and dulaglutide group to MetS, while at the end of reperfusion, O_2_^−^ values had significantly changed between all three groups as follows Exenatide > Dulaglutide > MetS (O_2_^−^ values (nmol/mL) at the end of stabilization and in the last minute of reperfusion, respectively—MetS—37.63, 35.09; Exenatide—26.44, 15.55; Dulaglutide—21.89, 21.01) (Figure 4C). When comparing values at the end of stabilization and at the end of reperfusion noticeable changes are in the group treated with exenatide (Figure 5C). Values of H_2_O_2_ were lower in both of the treated groups at the end of the stabilization period but increased during reperfusion with statistical significance in the dulaglutide group at 60 min of reperfusion compared to MetS (H_2_O_2_ values (nmol/mL) at the end of stabilization and in the last minute of reperfusion, respectively—MetS—36.41, 26.27; Exenatide—28.9, 33.86; Dulaglutide—25.51, 38.61) (Figure 4D). Changes between the stabilization point and the last minute of reperfusion were observed in the MetS group where we had markedly lower values of hydrogen peroxide, and the opposite was observed in the dulaglutide group where we had higher values (Figure 5D).

### 2.5. Effects of GLP-1RAs on Heart Morphology

The cross-section area of the cardiomyocytes was decreased in the exenatide group by 11% and in the dulaglutide group by 18% compared to the MetS group. In the histological photos of the HE staining, in the MetS group, we observed inflammation, wavy fibers, and cardiac muscle cells hypertrophy. In the groups of animals that were treated with exenatide and dulaglutide, we observed less pronounced inflammation compared to the MetS group and hypertrophy of individual muscle cells. The presence of wavy fibers was not observed (Figure 6).

In all experimental groups, there was a reduction in collagen levels when compared to the rats with MetS that did not receive any treatment. Specifically, the group treated with exenatide showed a 53% decrease in collagen content, while the dulaglutide group experienced a 68% reduction relative to the MetS group. The presence of collagen fibers, indicated by red deposits in Picro-sirius red staining shown in Figure 7, further supports these findings. Image analysis confirmed that treatment with both exenatide and dulaglutide effectively reduced the total collagen content in the heart, compared to the untreated MetS group (Figure 7).

## 3. Discussion

Given that MetS represents a complex combination of disorders including visceral obesity, insulin resistance, glucose intolerance, dyslipidemia, and hypertension, risk of CVD in these patients is multiplied. Despite the development in therapeutic approaches in treatment of both CVD and MetS, the effects are still limited, with a lack of understanding of the pathogenetic mechanisms. GLP-1RAs have several advantages in the treatment of MetS, and, therefore, the prevention of CVD, including reduction in glycemia, weight loss, and improvement in insulin tolerance [21,22,23]. Thus, the aim of this study was to assess and compare the effects of two GLP-1RAs, exenatide and dulaglutide, on in vivo heart function, cardiodynamic parameters of isolated hearts of MetS rats, systemic and cardiac oxidative stress biomarkers, and heart histomorphology in the experimental model of IRI.

Both applied drugs significantly reduced glycemia after 3 weeks of treatment, as well as after 6 weeks, at the end of the experimental protocol (Figure 1B). Values of glycemia during OGTT were significantly lower in both groups treated with GLP-1RAs compared to the MetS group, in all points of measurement, but there was no difference between the exenatide and dulaglutide (Figure 1C). A recent study conducted on human patients suffering from MetS showed that GLP-1RAs significantly improved the glycemic profile [24]. Body weight increased significantly in all groups, but the increase in body weight in the GLP-1RA-treated groups was less (Figure 1A). This result corresponds with a previously mentioned study due to a modest effect on body composition [24]. Both exenatide and dulaglutide reduced blood pressure, but only dulaglutide induced a statistically significant decrease (Figure 1E). It was shown in clinical trials that dulaglutide has an antihypertensive effect either as monotherapy or in combination with SGLT2 inhibitors [25,26]. On the other hand, a clinical study in which the effect of exenatide on blood pressure in type 2 diabetes (T2D) suffering patients was monitored showed that there was a trend of reduction without statistical significance, which is correlated with the results of this study [27,28]. One of the antihypertensive mechanisms of GLP-1RAs may involve an increase in atrium natriuretic peptide (ANP) [29]. Due to the wide distribution of GLP-1 receptors in the central nervous system, including regions involved in blood pressure control like hypothalamus and brainstem, it was shown that their activation caused a decrease in blood pressure [30]. Furthermore, distribution of GLP-1 receptors in vasculature, their activation and consequent increase in production of cAMP and improved activation of endothelial nitric oxide synthase (eNOS) participate in reduction in blood pressure [30].

Both applied GLP-1RA groups improved their echocardiographic parameters, but dulaglutide appeared to be more effective compared to the exenatide (Table 1). Interventricular septal wall thickness at end-systole and end-diastole (IVSs and IVSd), left-ventricular posterior wall thickness at end-systole and end-diastole (LVPWs and LVPWd), and left-ventricular internal diameter at end-systole (LVIDs) were deceased by both exenatide and dulaglutide, but the ejection fraction was increased only by dulaglutide (Table 1). Recent investigations showed that GLP-1RAs may prevent the development of heart failure and mortality related to T2D [31,32]. In a study dealing with the potential cardioprotective effects of GLP-1RAs on cardiomyocytes from MetS rats, liraglutide improved calcium homeostasis, reduced mitochondrial dysfunction and mitigated electrical abnormalities [33]. As such, GLP-1RAs induced protective mechanisms possibly underlining the prevention of heart dysfunction in MetS.

Ex vivo assessment of heart-function confirmed the protective potential of GLP-1RAs (Figure 2 and Figure 3). In the MetS group, parameters of cardiac contractility, dp/dt max, and dp/dt min, were significantly lower compared to the exenatide- and dulaglutide-treated group (Figure 2A,B). Furthermore, these parameters in the MetS group were significantly reduced by IRI, while in the exenatide- and dulaglutide-treated group dp/dt max and dp/dt min values in the last minute of reperfusion were similar to the initial values (Figure 3A,B). Similarly, SLVP was improved in the GLP-1RA-treated groups, while DLVP did not differ significantly between the groups (Figure 2C,D and Figure 3C,D). Both exenatide and dulaglutide improved the recovery of HR and CF after IRI (Figure 2E,F and Figure 3E,F). Several studies showed the cardioprotective effects of exenatide (or its structural analogue exendin-4) in acute application prior to IRI [34,35]. There is a lack of data related to examining the effects of dulaglutide in the experimental models of cardiac IRI, but we can assume that it shares a similar mechanisms of action. Those mechanisms appear to be pleiotropic including improvement of mitochondrial function, calcium regulation, and alleviation of electrical disturbances [33,34]. Electrophysiological studies reveal that GLP-1RAs may ameliorate the risk of arrhythmias in ischemic conditions by modulating the action potential duration and enhancing cardiac rhythm stability, likely through PKA-mediated ion channel activity [36]. A recent study showed that dulaglutide, beside the improvement in glucoregulation, efficiently prevented cardiac remodeling and heart dysfunction in T2D mice [32]. In a clinical trial dealing with the effects of dulaglutide on T2D patients, dulaglutide increased activity and the number of endothelial progenitor cells and amount of nitric oxide (NO). Also, reductions in the C-reactive protein (CRP), TNF-α, and IL-6 further illustrate the anti-inflammatory properties of GLP-1RAs, which play a crucial role in mitigating ischemia–reperfusion injury and endothelial dysfunction [37]. The anti-inflammatory effect of GLP-1RAs comes from a reduction in pro-inflammatory cytokines, primarily by inhibiting the phosphorylation and nuclear movement of NF-κB. Another mechanism involves an increase in IL-10 in the heart [38,39]. The main protective action of GLP-1RAs in ischemic myocardium involves cAMP-induced protein kinase A (PKA) activation, but some studies suggested some GLP-1-receptor-independent mechanisms of GLP-1RA cardioprotection [40]. GLP-1-receptor-independent cardioprotection may be mediated through the NO/cGMP-dependent pathway given that we also measured the increase in NO_2_^−^ values, as a marker of NO production, in coronary venous effluent (Figure 4C and Figure 5C) [41]. A recent study showed that exenatide and glucagon have additive cardioprotective effects in the postconditioned isolated rat heart [42]. Co-administration of exenatide and glucagon elicited better recovery of the contractile function after ischemia in the isolated hearts. Analyzing the effects of exentide and dulaglutide in our research on in vivo and ex vivo heart function, it seems that dulaglutide exhibits a somewhat more pronounced cardioprotective effect. A possible explanation (besides receptor-independent molecular mechanisms) lies in their formulation and pharmacokinetic profiles, dosing frequency which affects their onset of action, and even individual variety and pharmacogenetics [27,43,44]

Due to the known link between oxidative stress, MetS, and IRI, we aimed to assess how GLP-1RAs affect coronary outflow of oxidative stress biomarkers. For both the investigated GLP-1RA groups, exenatide and dulaglutide, the effect of dulaglutide was more pronounced, and there were increased values of NO_2_^−^ in the coronary venous effluent (Figure 4B). Values of NO_2_^−^ were higher even during the stabilization period compared to MetS, but they were slightly increased during reperfusion (Figure 4B). Values of H_2_O_2_ were lower in GLP-1RA-treated hearts at the end of the stabilization period but significantly increased during reperfusion (Figure 4D and Figure 5D). O_2_^−^ values were significantly lower in GLP-1RA-treated hearts in the last minute of reperfusion compared to the MetS group, and further decreased during reperfusion (Figure 4C and Figure 5C). TBARS as an index of lipid peroxidation did not changed significantly (Figure 4A and Figure 5A). Due to increased values of H_2_O_2_ and decreased values of O_2_^−^ levels, we could assume that GLP-1RAs induced the increased endothelial and cardiomyocyte activity of SOD. This assumption is in line with previous findings that exenatide increases SOD levels and preserves the mitochondrial membrane potential in an in vitro model of hypoxia/reoxygenation of H9c2 cells [45]. Another study showed that exendin-4 (exenatide analogue) mitigated cardiac oxidative stress in T2D mice by suppressing NADPH oxidase 4 followed by increasing the activity of SOD-1 and glutathione peroxidase (GPx) [46]. A recent study on human ventricular cardiomyocyte cell lines (AC16) showed that exendin-4 reduced oxidative stress, improved the mitochondrial network, and prevented apoptosis [47]. It was also shown that another GLP-1RA, liraglutide, improves oxidative damage in human umbilical vein endothelial cells (HUVECs) induced by TNF-α [48]. Liraglutide inhibited NADPH oxidase activity and increased the protein levels of SOD, CAT, and GPx. A recent study also showed that liraglutide-mitigated isoprenaline-induced myocardial injury in rats caused by a reduction in oxidative stress, improvement of CAT and SOD activity and markers of cardiac damage such as high-sensitivity troponin I, aspartate aminotransferase, alanine aminotransferase [49]. Moreover, GLP-1RAs appear to improve cardiac metabolism through enhanced glucose uptake, increased fatty acid oxidation, and improved mitochondrial function, contributing to better energy utilization and reduced oxidative stress in cardiac tissues [50]. Although, there are limited data regarding the effects of dulaglutide on cardiac oxidative stress, it was shown that dulaglutide suppressed lipopolysaccharide (LPS)-induced oxidative stress in H9c2 myocardial cells and increased the myocardial content of GSH [51]. The results of this study indicated a similar antioxidant capacity of both applied GLP-1RAs, but due to the effects on O_2_^−^ levels (Figure 4C and Figure 5C), exenatide had a more pronounced antioxidant potential. Altogether, our results and those of others suggest the importance of the antioxidative activity of GLP-1RAs in cardioprotection.

A histomorphological analysis showed significant improvement in GLP-1RA-treated groups compared to MetS (Figure 6). Furthermore, the collagen content was significantly reduced by both applied GLP-1RAs, 53% by exenatide and 68% by dulaglutide (Figure 7). Similar results were accomplished by liraglutide in the experimental model of MI [52]. Collagen fiber Masson staining showed improved myocardial fibrosis after MI and indicated the role of reduction in the connective tissue growth factor (CTGF) in this effect. Similarly, liraglutide prevented the proliferation of myofibroblasts and reduced the cardiac collagen in the experimental model of myocardial fibrosis induced by angiotensin II infusion [53]. Recent research showed the antifibrotic effect of exenatide in the experimental model of high-fructose-induced MetS [54]. Exenatide induced a decrease in asymmetric dimethylarginine (ADMA), increased cardiac NO and reduced cardiac collagen content. Refaat et al. demonstrated that the activation of GLP-1 receptors leads to a reduction in myocardial fibrosis by inhibiting the TGF-β/Smad3 pathway [55]. Additionally, these agents increase the expression of matrix metalloproteinases (MMP), such as MMP-9, facilitating the degradation of excessive collagen in fibrotic tissues, thereby counteracting the fibrotic remodeling of the heart [56]. The modulation of fibroblast activity, preventing their differentiation into myofibroblasts, further exemplifies the antifibrotic actions of GLP-1RAs, potentially mediated through cAMP signaling pathways [57,58]. There is a lack of data regarding the effects of dulaglutide in cardiac fibrosis, but it is shown that is can prevent the experimental renal fibrosis induced by high-fructose intake [59]. Also, Matsubara et al. showed that GLP-1 (fusion of GLP-1 to human transferrin) primarily acts by mitigating the damage caused by reperfusion, rather than enhancing the heart muscle’s ability to withstand ischemia. The protective effect on the heart is attributed, in part, to a decrease in the apoptosis of heart cells triggered by reperfusion [60].

The limitations of our study are mainly reflected in the lack of dissection of molecular mechanisms that mediate a beneficial effect of both applied GLP-1RAs, exenatide and dulaglutide. Furthermore, it would be of interest to include more GLP-1RAs and to assess whether they achieve cardioprotection through the same mechanisms. The lack of translation to the human respondents also could be recognized as a limitation of the study.

Future research should focus on unraveling the specific molecular mechanisms behind the cardioprotective effects of exenatide and dulaglutide. In that sense, the effects of GLP-1RAs on processes like inflammation, cell death, and cell preservation within the heart should be included in future research. Additionally, extending the duration of treatment beyond six weeks and broadening the scope of research to include different dosages and other GLP-1RAs might prove beneficial. To understand the estrogen cardioprotective effects in IRI means examining the potential variations in efficacy due to gender differences [61]. Moreover, investigating the effects of GLP-1RA therapy across different MetS models, including those based on genetic predispositions and dietary factors, will aid in determining the consistency of the therapy’s effectiveness across various scenarios.

It would also be interesting to add some other medications as a combination with GLP-1RAs, which are used in the treatment of metabolic syndrome. For instance, statins have shown cardioprotective effects in IRI; additionally, combining atorvastatin with exenatide has been found to counteract the reduction in insulin levels and the increase in LDL-R expression in human pancreatic beta cells caused by atorvastatin [62,63].

Furthermore, an important direction in the future could be the translation of basic research to findings in a human population.

Altogether, these results suggest there is strong cardioprotective potential for GLP-1RAs, exenatide and dulaglutide, in an experimental model of MetS, where the protective abilities of dulaglutide are even more pronounced. One of the main mechanisms of action of GLP-1RAs in cardioprotection appears to be the antioxidative effect. The reduction in oxidative stress and probable improvement in antioxidative capacity in the heart achieved by GLP-1RAs may be the target of interest in cardioprotection and reduction in the mortality of CVD in MetS and diabetes-suffering patients. Furthermore, the observed reduction in fibrosis appears to significantly support reducing the consequences of the IRI of the myocardium, and thus real supportive therapy after MI. These insights not only underscore the therapeutic value of GLP-1RAs in cardiovascular protection but also pave the way for future research and offer new hope in the battle against cardiovascular disease.

## 4. Materials and Methods

### 4.1. Animals and Experimental Design

MetS was induced in 4-week-old Wistar albino male rats obtained from the Medical Military Academy Belgrade. Rats were housed under controlled environmental conditions: temperature (22 ± 2), humidity, and light/dark cycle (12 h/12 h), with free access to food and water. The animal study protocol was approved by the Ethical Committee for Laboratory Animal Welfare of the Faculty of Medical Sciences, University of Kragujevac, Serbia (approval No: 01-11876/4, date: 11 October 2019). All research procedures described in the study were carried out in accordance with the European Directive for the welfare of laboratory No 86/609/EEC and the principles of Good Laboratory Practice. The protocol for the induction of MetS was a high-fat diet (HFD) applied for 4 weeks, followed by intraperitoneal injection of streptozotocin (STZ) in a dose of 25 mg/kg after 12 h of fasting [64]. In total, 72 h after the administration of STZ, fasting blood glucose and blood pressure are measured. MetS was confirmed if the fasting blood glucose level was over 7 mmol/L and the blood pressure value was over 130/90 mmHg. Rats with confirmed MetS (24 in total) were randomly divided into three groups (eight in each group):MetS group—control group, rats treated with saline subcutaneously (NaCl 0.9%) in a dose of 0.5 mL per day for 6 weeks;Exenatide group—rats treated with exenatide subcutaneously in a dose of 5 µg/kg per day for 6 weeks [65];Dulaglutide group—rats treated with dulaglutide subcutaneously in a dose of 0.6 mg/kg/twice a week for 6 weeks [66].

### 4.2. Physiological Parameters (Weight, Glycemia, and Insulin Values during the Oral Glucose Tolerance Test

Body weight and glycemia were measured after (for) the confirmation of MetS, after three weeks of drug application, and at the end of the experimental protocol (before sacrifice). Glycemia was measured after twelve-hour fasting using an Accu-Chek glucometer (Roche Diagnostics, Indianapolis, IN, USA) with its comparing strips. Also, an oral glucose tolerance test (OGTT) was performed at the end of the experimental protocol (before sacrifice) after twelve-hour fasting. Blood was sampled by puncturing the tail vein. Immediately after the determination of fasting glycemia (0), glucose in a dose of 2 g/kg body weight was administered using oral gavage. Glycemia was measured at 0, 30, 60, 120, and 180 min, while insulin levels were measured at 0 and 180 min using the enzyme-linked immunosorbent assay (ELISA) method [64,67].

### 4.3. In Vivo and Ex Vivo Examination of Cardiac Function

Blood pressure was measured at the beginning and at six week of the experimental protocol using a tail-cuff method (Rat Tail Cuff Method Blood Pressure Systems (MRBP-R), IITC Life Science Inc., Los Angeles, CA, USA). Rats were placed in a chamber for approximately 30 min, 3 days before measurement, in order to avoid the possible effects of stress oscillations on blood pressure results.

Transthoracic echocardiography was performed at the end of the experimental protocol, after the sixth week of application of GLP-1RAs. Prior to the procedure, the rats were anaesthetized with an intraperitoneal application of ketamine and xylazin (10 mg/kg and 5 mg/kg, respectively). In the M-mode and from the parasternal long-axis view (PLAX), the following parameters were measured: the left-ventricle (LV) posterior wall thickness at end-diastole (LVPWd) and end-systole (LVPWs), interventricular septal wall thickness at end-diastole (IVSd) and end-systole (IVSs), left-ventricle internal dimension at end-diastole (LVIDd) and end-systole (LVIDs), and the fractional shortening (FS) percentage. In addition, for calculation of the ejection fraction, we used the Teicholz formula [68]. Echocardiograms were performed using a Hewlett-Packard Sonos 5500 (Andover, MA, USA) sector scanner equipped with a 15.0-MHz phased-array transducer.

After transthoracic echocardiography, we used the Langendorff technique of an isolated mammalian heart for the ex vivo examination of heart function. While they were still under anesthesia, rats were sacrificed with decapitation. After a prompt thoracotomy, the hearts were removed and submerged into the ice-cold saline. The hearts were attached to Langendorff apparatus via aortic cannula to provide retrograde perfusion under constant coronary pressure (CPP = 70 cmH_2_O). Hearts were perfused with Krebs–Henseleit solution (NaCl 118 mmol/L, KCl 4.7 mmol/L, CaCl_2_ × 2H_2_O 2.5 mmol/L, MgSO_4_ × 7H_2_O 1.7 mmol/L, NaHCO_3_ 25 mmol/L, KH_2_PO_4_ 1.2 mmol/L, glucose 11 mmol/L, pyruvate 2 mmol/L), gassed with 95% O_2_ and 5% CO_2_, and heated to 37 °C (pH  7.4). After making an incision on the left atrium and rupture of the mitral valves, a transducer (BS473-0184, Experimetria Ltd., Budapest, Hungary) was placed in the left ventricle (LV) for continuously monitoring the following parameters of myocardial function:The maximum rate of pressure development in the left ventricle (dp/dt max);The minimum rate of pressure development in the left ventricle (dp/dt min);The systolic left ventricular pressure (SLVP);The diastolic left ventricular pressure (DLVP);The heart rate (HR).

Coronary flow (CF) was measured flowmetrically.

After the stabilization period, which is determined using the three same values of flowmetrically measured coronary flow (CF), the heart underwent global ischemia (perfusion through the heart was totally stopped) for 30 min, followed by 60 min of reperfusion. The cardiodynamic parameters and coronary flow were measured in the next point of interest: the last minute of stabilization, 1st, 3rd, 5th, 10th, 15th, 30th, 45th, 60th minute of reperfusion. At each point of interest, CF was measured and the coronary venous effluent was collected for further analysis of the oxidative stress biomarkers [68].

### 4.4. Redox Status

The evaluation of pro-oxidative biomarkers was performed in coronary venous effluent. All oxidative stress biomarkers were determined spectrophotometrically.

We measured the following pro-oxidative markers:The index of lipid peroxidation, measured as thiobarbituric acid reactive substances (TBARS);The level of nitrite (NO_2_^−^);The level of superoxide anion radical (O_2_^−^);The level of hydrogen peroxide (H_2_O_2_).

#### 4.4.1. TBARS Determination

Thiobarbituric acid reactive substances were used for the estimation of the lipid peroxidation level in the plasma and coronary venous effluent. Thiobarbituric acid (1%) in 0.05 NaOH was incubated with coronary effluent or plasma samples at 100 °C for 15 min and then read at a wavelength of λ = 530 nm. As a blank, we used a Krebs–Henseleit solution (method described in detail in [69,70]).

#### 4.4.2. Nitrite Determination

NO_2_^−^ measurement was used to estimate the nitrogen monoxide (NO) production. Green’s method was used for the determination of NO_2_^−^ values. Freshly prepared Griess’s reagent was used for the spectrophotometrically determination of NO_2_^−^ levels at a wavelength of λ = 530 nm. The nitrite levels were calculated using sodium nitrite as the standard (method described in detail in [69,70]).

#### 4.4.3. Superoxide Anion Radical Determination

Estimation of the O_2_^–^ in plasma and coronary venous effluent samples is based on the reaction of nitro blue tetrazolium (NBT) in tris (hydroxymethyl) aminomethane (TRIS)-buffer with O_2_^–^ and the formation of nitro formazan blue. The measurement was conducted at a wavelength of λ = 530 nm. Krebs–Henseleit solution was used as a blank probe for the O_2_^–^ determination in coronary venous effluent (method described in detail in [69,70]).

#### 4.4.4. Hydrogen Peroxide Determination

The estimation of H_2_O_2_ was based on the oxidation of phenol red by H_2_O_2_ in a reaction catalyzed by the enzyme horseradish peroxidase (HRPO). Krebs–Henseleit solution was used as a blank probe for H_2_O_2_ determination. The measurement was made at a wavelength of λ = 610 nm (method described in detail in [69,70]).

### 4.5. Histological Analysis

After the ex vivo experimental protocol on Langendorff apparatus, the hearts were detached and immersed into 4% paraformaldehyde solution for 24 h. After fixation, the hearts were dehydrated (increasing concentrations of ethanol), enlightened in xylol and embedded in Histowax^®^ (Histolab Product AB, Göteborg, Sweden). Molded blocks of heart samples were cut on a rotational microtome (RM 2125RT Leica Microsystems, Wetzlar, Germany) into 5 µm thick sections. Tissue sections were stained with hematoxylin–eosin for the visualization of tissue structures, and with picro-sirius red for the detection of the collagen content. The picro-sirius red-staining method results in dark-brown- to black-stained nuclei, yellowish muscle cell cytoplasm, and redish stains of collagen. Images of the tissue sections were obtained using a digital camera connected to an Olympus BX51 microscope. For morphometric analysis, we utilized the calibrated Axiovision software 4.8 (Carl Zeiss, White Plains, NY, USA) and Image Pro-Plus 7.0 (Media Cybernetics, Rockville, MD, USA), following the methodology we previously outlined [71]. We conducted the measurement of the cardiomyocyte cross-section area (as part of the morphometric analysis) on a minimum of 100 cells for each animal. The results for the collagen content were expressed as a percentage [71,72,73].

### 4.6. Statistical Analysis

All data were analysed using and GraphPad Prism 8 (Version for Windows, GraphPad Software, La Jolla, CA, USA). The results are expressed as means ± standard deviation of the mean (SD). To assess the normality of the data distribution, the Shapiro–Wilk test was employed. After confirmation of normal parameter distribution, independent samples *t*-test (parametric) and Mann–Whitney U-test (nonparametric) as well as one-way ANOVA alongside Turkey’s post hoc test and Kruskal–Wallis were used to assess the difference in estimated variables between the groups. A *p* value <0.05 was regarded as statistically significant.

## Figures and Tables

**Figure 1 pharmaceuticals-17-00525-f001:**
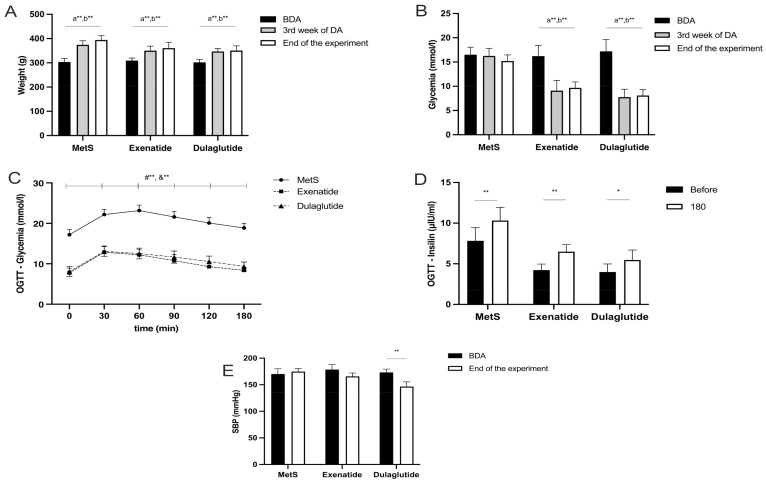
(**A**) Effects of GLP-1RA on weight, (**B**) effect of GLP-RA on glycemia levels, (**C**) effect of GLP-1RA on glucose level during OGTT, (**D**) effect of GLP-1RA on insulin level during OGTT, (**E**) effect of GLP-1RA on systolic blood pressure. Values are presented as mean ± SD. Statistical significance at the level * *p* < 0.05, ** *p* < 0.01. MetS: metabolic syndrome group. Exenatide: group of rats treated with exenatide. Dulaglutide; group of rats treated with dulaglutide. BDA: before drug administration. 3rd week of DA: third week of drug administration. Before: before OGTT (glucose administration) a, #—MetS vs. Exenatide; b, &—MetS vs. Dulaglutide.

**Figure 2 pharmaceuticals-17-00525-f002:**
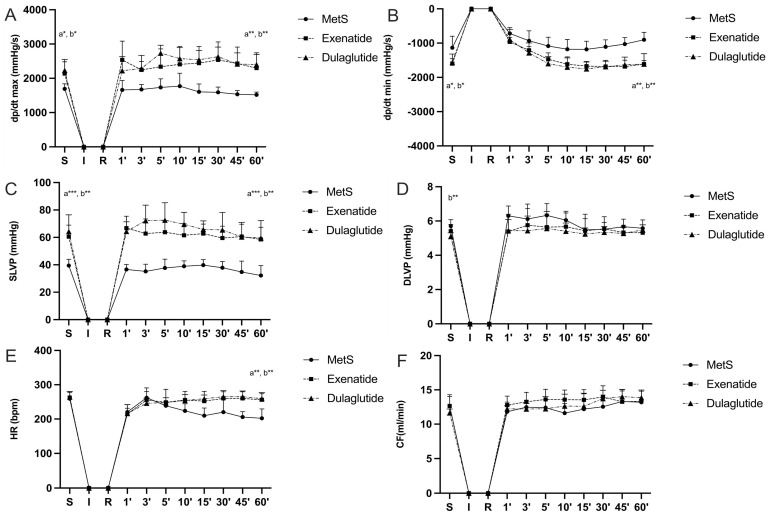
Effect of GLP-1RAs on cardiodynamic parameters: (**A**) dp/dt max—the maximum rate of LV pressure development, (**B**) dp/dt min—minimum rate of LV pressure development, (**C**) SLVP—systolic LV pressure, (**D**) DLVP—diastolic LV pressure, (**E**) HR—heart rate, and (**F**) CF—coronary flow. Values are presented as mean ± SD. Statistical significance at the level * *p* < 0.05, ** *p* < 0.01, *** *p* < 0.001. MetS: metabolic syndrome group; exenatide: group of rats treated with exenatide; dulaglutide; group of rats treated with dulaglutide. a—MetS vs. Exenatide; b—MetS vs. Dulaglutide. S—stabilization, I—ischemia, R—reperfusion.

**Figure 3 pharmaceuticals-17-00525-f003:**
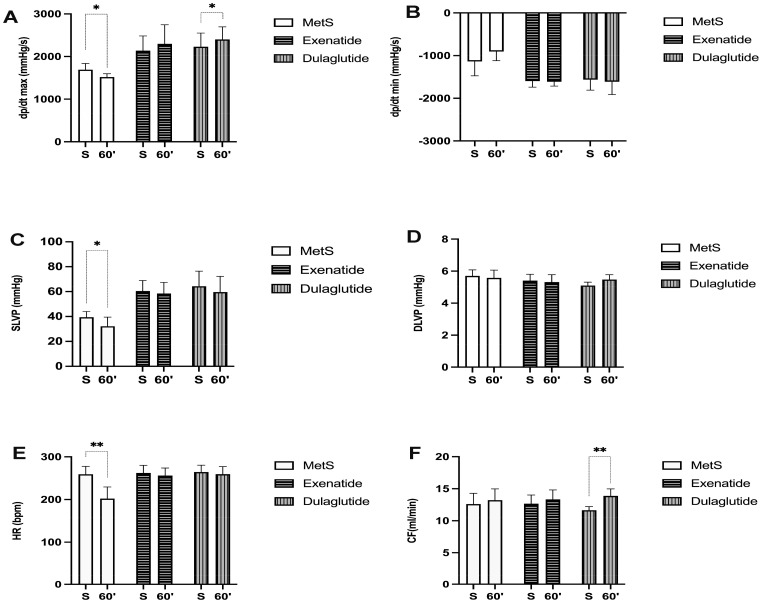
Effect of GLP-1RAs on cardiodynamic parameters: (**A**) dp/dt max—the maximum rate of LV pressure development, (**B**) dp/dt min—minimum rate of LV pressure development, (**C**) SLVP—systolic LV pressure, (**D**) DLVP—diastolic LV pressure, (**E**) HR—heart rate, and (**F**) CF—coronary flow. Values are presented as mean ± SD, statistical significance at the level * *p* < 0.05, ** *p* < 0.01. MetS: metabolic syndrome group; exenatide: group of rats treated with exenatide; dulaglutide: group of rats treated with dulaglutide; S—stabilization period; 60′—60 (last) minute of reperfusion.

**Figure 4 pharmaceuticals-17-00525-f004:**
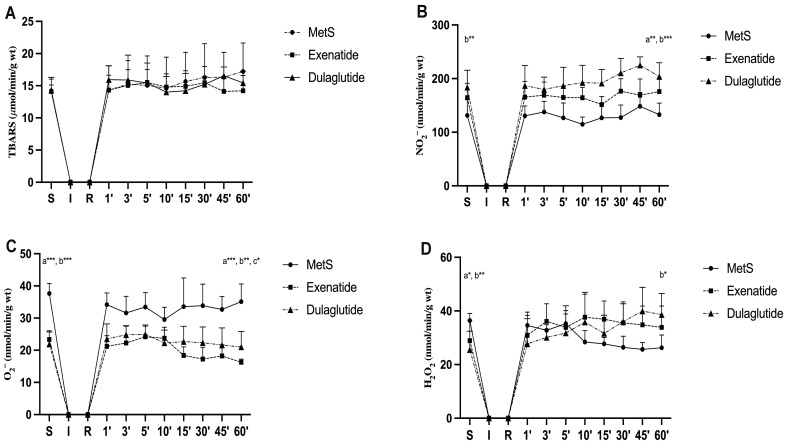
Effect of GLP-1RAs on pro-oxidants in coronary venous effluent: (**A**) TBARS—index of lipid peroxidation, (**B**) NO_2_^−^—nitrite, (**C**) O_2_^−^—superoxide anion radical and, (**D**) H_2_O_2_—hydrogen peroxide. Values are presented as mean ± SD. Statistical significance at the level * *p* < 0.05, ** *p* < 0.01, *** *p* < 0.001. MetS: metabolic syndrome group; exenatide: group of rats treated with exenatide; dulaglutide: group of rats treated with dulaglutide;. a—MetS vs. Exenatide; b—MetS vs. Dulaglutide; c—Exenatide vs. Dulaglutide. S—stabilization, I—ischemia, R—reperfusion.

**Figure 5 pharmaceuticals-17-00525-f005:**
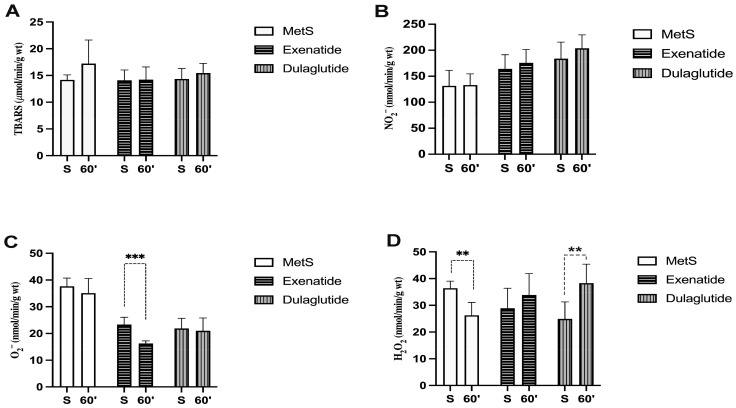
Effect of GLP-1RAs on pro-oxidants in coronary venous effluent: (**A**) TBARS—index of lipid peroxidation, (**B**) NO_2_^−^—nitrite, (**C**) O_2_^−^—superoxide anion radical and, (**D**) H_2_O_2_—hydrogen peroxide. Values are presented as mean ± SD, statistical significance at the level ** *p* < 0.01, *** *p* < 0.001. MetS: metabolic syndrome group; exenatide: group of rats treated with exenatide; dulaglutide: group of rats treated with dulaglutide; S—stabilization period; 60′—60 (last) minute of reperfusion.

**Figure 6 pharmaceuticals-17-00525-f006:**
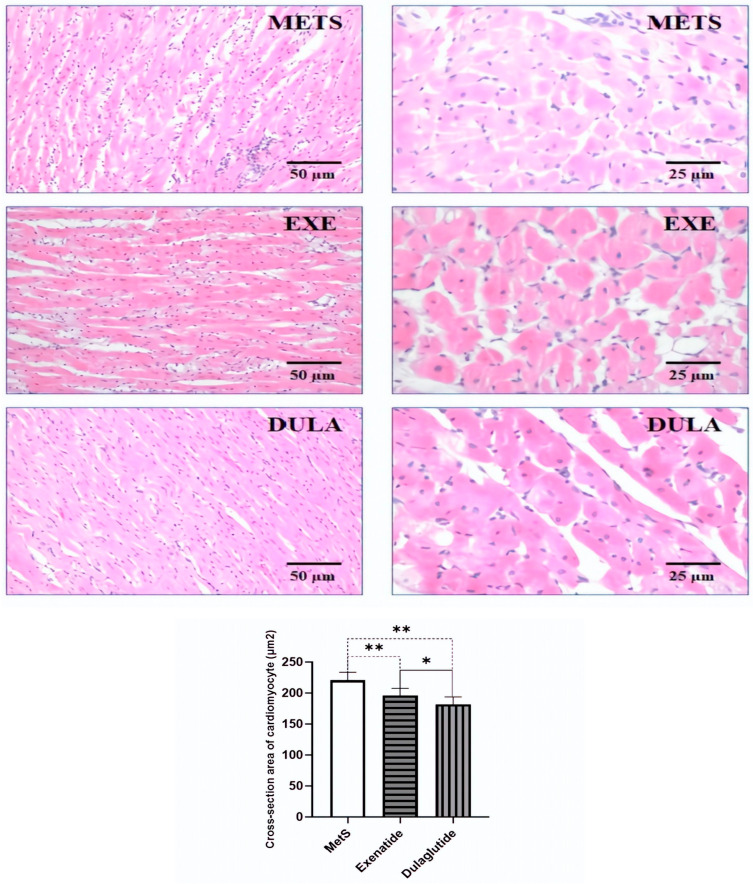
Representative micrograph of H&E staining of the cardiac muscle. Left row (objective magnification 20×, bar = 50 μm). Right row (objective magnification 40×, bar = 25 μm). METS: metabolic syndrome group; EXE: group of rats treated with exenatide; DULA: group of rats treated with dulaglutide. Graph represents values of the cross-section area of the cardiomyocyte of all groups. All values are the mean value ± SD. Statistical significance at the level * *p* < 0.05, ** *p* < 0.01 vs. the adequate group, connected by a horizontal line.

**Figure 7 pharmaceuticals-17-00525-f007:**
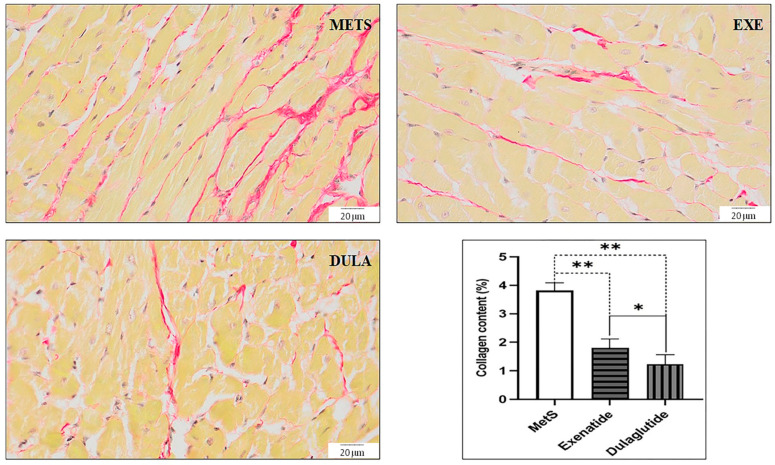
Representative micrographs of Picrosirius red-stained rat cardiac muscle sections (objective magnification 40×, bar = 20 μm; collagen fibers). METS: metabolic syndrome group; EXE: group of rats treated with exenatide; DULA: group of rats treated with dulaglutide. Graph represent quantification of collagen content (%) for all groups. All values are the mean value ± SD, statistical significance at the level * *p* < 0.05, ** *p* < 0.01 vs. the adequate group, connected by a horizontal line.

**Table 1 pharmaceuticals-17-00525-t001:** Effects of GLP-1RAs on echocardiographic parameters: interventricular septal wall thickness at end-systole and end-diastole (IVSs and IVSd), left-ventricular internal diameter at end-systole and end-diastole (LVIDs and LVIDd), left-ventricular posterior wall thickness at endsystole and end-diastole (LVPWs and LVPWd), fractional shortening (FS), and ejection fraction (EF). MetS: metabolic syndrome group. Exenatide: group of rats treated with exenatide. Dulaglutide: group of rats treated with dulaglutide. Values are presented as mean ± SD. Statistical significance at the level of *p* < 0.05. ∞-MetS vs. Exenatide; #-MetS vs. Dulaglutide; §-Exenatide vs. Dulaglutide.

	MetS	Exenatide	Dulaglutide
IVSd (cm)	0.332 ± 0.079	0.140 ± 0.031∞	0.140 ± 0.020#
LVIDd (cm)	0.675 ± 0.050	0.754 ± 0.024	0.676 ± 0.071
LVPWd (cm)	0.178 ± 0.008	0.157 ± 0.039∞	0.140 ± 0.005#§
IVSs (cm)	0.214 ± 0.019	0.173 ± 0.029∞	0.147 ± 0.023#§
LVIDs (cm)	0.485 ± 0.056	0.415 ± 0.018∞	0.341 ± 0.039#
LVPWs (cm)	0.216 ± 0.021	0.184 ± 0.019∞	0.170 ± 0.041#
FS (%)	42.075 ± 3.767	41.367 ± 9.136	47.725 ± 5.291
EF (%)	78.586 ± 4.086	81.312 ± 1.283	85.248 ± 3.301#§

## Data Availability

The data presented in this study are available on request from the corresponding author.

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
