# Peer review of "Effect of GLP-1 Receptor Agonist on Ischemia Reperfusion Injury in Rats with Metabolic Syndrome"

_pharmaceuticals, 2024, doi:10.3390/ph17040525_

Round 1

Reviewer 1 Report

Comments and Suggestions for Authors

Title ;  Effect of GLP-1 receptor agonist on ischemia reperfusion injury  in rats with metabolic syndrome

This study aims to assess the influence of administering two GLP-1 analogs on metabolic parameters in rats with metabolic syndrome. The findings indicate that these two analogs effectively ameliorate metabolic syndrome. However, significant revisions are necessary for this manuscript. Attached are the suggested improvements.

1.    The abstract lacks specificity; it should detail the measured parameters and indices and a comparative analysis between treated and untreated rats, including numerical values and statistical analysis.

2.    The introduction should commence by highlighting the significance of the study's objective, such as the prevalence of myocardial ischemia and reperfusion cases and the associated complications.

3.    In the results section, include numerical values and percentages for all measured indices to enhance clarity.

4.    Ensure all abbreviations are expanded upon their first mention, with the abbreviation provided alongside in the abstract.

5.    Question the rationale behind the selected doses mentioned in lines 423-425, as they may appear inadequate.

Comments on the Quality of English Language

Title ;  Effect of GLP-1 receptor agonist on ischemia reperfusion injury  in rats with metabolic syndrome

This study aims to assess the influence of administering two GLP-1 analogs on metabolic parameters in rats with metabolic syndrome. The findings indicate that these two analogs effectively ameliorate metabolic syndrome. However, significant revisions are necessary for this manuscript. Attached are the suggested improvements.

1.    The abstract lacks specificity; it should detail the measured parameters and indices and a comparative analysis between treated and untreated rats, including numerical values and statistical analysis.

2.    The introduction should commence by highlighting the significance of the study's objective, such as the prevalence of myocardial ischemia and reperfusion cases and the associated complications.

3.    In the results section, include numerical values and percentages for all measured indices to enhance clarity.

4.    Ensure all abbreviations are expanded upon their first mention, with the abbreviation provided alongside in the abstract.

5.    Question the rationale behind the selected doses mentioned in lines 423-425, as they may appear inadequate.

Author Response

Open Review

Quality of English Language

( ) I am not qualified to assess the quality of English in this paper
( ) English very difficult to understand/incomprehensible
( ) Extensive editing of English language required
(x) Moderate editing of English language required
( ) Minor editing of English language required
( ) English language fine. No issues detected

Yes

Can be improved

Must be improved

Not applicable

Does the introduction provide sufficient background and include all relevant references?

( )

( )

(x)

( )

Are all the cited references relevant to the research?

( )

(x)

( )

( )

Is the research design appropriate?

(x)

( )

( )

( )

Are the methods adequately described?

(x)

( )

( )

( )

Are the results clearly presented?

( )

( )

(x)

( )

Are the conclusions supported by the results?

( )

( )

(x)

( )

Comments and Suggestions for Authors

Title ;  Effect of GLP-1 receptor agonist on ischemia reperfusion injury  in rats with metabolic syndrome

This study aims to assess the influence of administering two GLP-1 analogs on metabolic parameters in rats with metabolic syndrome. The findings indicate that these two analogs effectively ameliorate metabolic syndrome. However, significant revisions are necessary for this manuscript. Attached are the suggested improvements.

Thank you for your comments. We corrected the manuscript according to your suggestions, improved the presentation of the results and specified the conclusion.

  1. The abstract lacks specificity; it should detail the measured parameters and indices and a comparative analysis between treated and untreated rats, including numerical values and statistical analysis.

We adjusted the abstract according to your suggestions.

  1. The introduction should commence by highlighting the significance of the study's objective, such as the prevalence of myocardial ischemia and reperfusion cases and the associated complications.

Thank you for your suggestion. The importance of finding new therapeutic approaches for myocardial ischemia due to high incidence of myocardial infarction and metabolic syndrome is already mentioned in the first paragraph of the introduction. We additionally highlighted the significance of study objective before the aim of the study – please see the lines 92-98.

  1. In the results section, include numerical values and percentages for all measured indices to enhance clarity.

We have added numerical values to the text explaining statistical significance. Please see the results section.

  1. Ensure all abbreviations are expanded upon their first mention, with the abbreviation provided alongside in the abstract.

Thank you for your suggestion. We checked the abbreviations throughout the text.

  1. Question the rationale behind the selected doses mentioned in lines 423-425, as they may appear inadequate.

Thank you for your suggestion. We corrected the dose for streptozotocin - 25 mg/kg (instead of 0.25 mg/kg) ,exenatide – 5 µg/kg (instead of 0.5 µg/kg) and dulaglutide – 0.6 mg/kg (instead of 0.3 mg/kg). We cited references for doses of GLP1-RAs. Please see the material and methods section.

Comments on the Quality of English Language

Please see the attachments. Answers to the Reviewer are marked in red. 

Reviewer 2 Report

Comments and Suggestions for Authors

The study aims to examine the effects of GLP-1 receptor agonists (GLP-1RA), specifically exenatide and dulaglutide, on heart function and redox balance in rats with induced Metabolic Syndrome (MetS) subject to ischemia-reperfusion injury. This investigation is significant due to the increased risk of myocardial infarction and complications associated with MetS. The following comments and suggestions may aid the authors enhancing the clarity, depth, and impact of their manuscript, contributing significantly to the existing literature on GLP-1 receptor agonists and cardiovascular health in metabolic syndrome contexts.

General points:

Suggestions for improvement:

  1. Expand the discussion section to include a deeper analysis of the comparative effectiveness of exenatide and dulaglutide.
  2. Introduce a limitations section discussing potential biases, the limitations of animal models, and the implications for human treatment.
  3. While the animal model is suitable for this study type, translating these findings to human clinical practice requires caution; a section discussing this would be beneficial. Consider including additional studies or a meta-analysis to strengthen the context and relevance of the findings to existing literature.
  4. Improve the overall readability of the manuscript by reducing jargon and simplifying complex sentences.
  5. Consider the addition of a "Future Directions" section to suggest how this research could evolve, and other related areas could be explored.
  6. Ensure consistency in terminology, particularly when referring to the study groups, treatments, and outcomes.

Introduction:

  1. Provide a clearer statement of the research gap your study aims to address. This will help set the context for your study's significance.
  2. Expand on the mechanism through which GLP-1 receptor agonists are hypothesized to affect ischemia-reperfusion injury, providing a stronger theoretical foundation for your research.

Methods:

  1. Detail any ethical considerations or animal welfare measures taken during the study to reassure the reader of its ethical standards.
  2. Include specifics about the randomization and blinding processes to enhance the reliability and validity of your results.

3.      Clarify the selection criteria for the animal model and justify the choice of doses for exenatide and dulaglutide.

Discussion:

  1. Dive deeper into how your findings align or contrast with existing literature, providing a more detailed analysis of potential reasons for any discrepancies.
  2. Discuss the potential implications of your findings for future clinical applications, including any possible limitations in extrapolating animal model findings to humans.
  3. Address the potential mechanisms behind the differential effects observed between exenatide and dulaglutide, hypothesizing why one might be more effective than the other.

Limitations:

  1. Acknowledge limitations related to the study design, the animal model used, and the applicability of results to human populations.
  2. Discuss how these limitations could impact the interpretation of your findings and suggest areas for future research to address these gaps.

Figures and Tables:

  1. Ensure all figures and tables are of high quality, clearly labelled, and referenced in the text. The black columns in figure 1 can not differentiated by the reader, please consider that in all figures.

General considerations on the language and grammar

Consistency in Terminology:

  • Inconsistent: "The study investigates GLP-1 receptor agonists in the first section, then examines the impacts of GLP-1RA drugs later."
  • Consistent: "The study investigates GLP-1 receptor agonists (GLP-1RAs) in the first section and continues to examine the impacts of GLP-1RAs throughout."

Sentence Structure:

  • Complex Sentence: "Given that metabolic syndrome increases the risk of cardiovascular disease, which is further complicated by factors such as obesity and insulin resistance, the study of GLP-1 receptor agonists becomes essential."
  • Simplified Sentences: "Metabolic syndrome increases the risk of cardiovascular disease. Factors such as obesity and insulin resistance complicate this condition. Therefore, studying GLP-1 receptor agonists is essential."

Grammar and Punctuation:

  • Incorrect: "After conducting the experiments, the rats were analysed, and their data was collected and reviewed."
  • Correct: "After conducting the experiments, we analyzed the rats and collected and reviewed their data."

Precision and Clarity:

  • Vague: "The drug had a good effect on the heart."
  • Precise: "The drug significantly reduced myocardial infarction size by 30%."

Style and Tone:

  • Informal: "The results were super interesting and showed we’re on the right track."
  • Formal: "The results were significant and indicated that the current research direction is promising."

Consistency in Formatting:

  • Inconsistent: "Table 1 shows... However, in Figure 2 we see..."
  • Consistent: "Table 1 shows... Similarly, Figure 2 illustrates..."

Proofreading:

  • Before: "The experiment showed that heart rate decreased in the control group."
  • After: "The experiment showed that the heart rate decreased in the control group."

By addressing these specific examples in your manuscript, you can enhance its overall quality and ensure it adheres to academic standards.

Comments on the Quality of English Language

The English language needs moderate revisions.

Author Response

Open Review

Quality of English Language

( ) I am not qualified to assess the quality of English in this paper
( ) English very difficult to understand/incomprehensible
( ) Extensive editing of English language required
(x) Moderate editing of English language required
( ) Minor editing of English language required
( ) English language fine. No issues detected

Yes

Can be improved

Must be improved

Not applicable

Does the introduction provide sufficient background and include all relevant references?

( )

(x)

( )

( )

Are all the cited references relevant to the research?

( )

(x)

( )

( )

Is the research design appropriate?

(x)

( )

( )

( )

Are the methods adequately described?

( )

(x)

( )

( )

Are the results clearly presented?

( )

(x)

( )

( )

Are the conclusions supported by the results?

( )

(x)

( )

( )

Comments and Suggestions for Authors

The study aims to examine the effects of GLP-1 receptor agonists (GLP-1RA), specifically exenatide and dulaglutide, on heart function and redox balance in rats with induced Metabolic Syndrome (MetS) subject to ischemia-reperfusion injury. This investigation is significant due to the increased risk of myocardial infarction and complications associated with MetS. The following comments and suggestions may aid the authors enhancing the clarity, depth, and impact of their manuscript, contributing significantly to the existing literature on GLP-1 receptor agonists and cardiovascular health in metabolic syndrome contexts.

General points:

Suggestions for improvement:

  1. Expand the discussion section to include a deeper analysis of the comparative effectiveness of exenatide and dulaglutide.

Thank you for your suggestion. The analysis regarding the comparison of the effects of dulaglutide and exenatide on glycemia and blood pressure are already done in discussion – please see lines 373-394. We added the potential mechanisms regarding the cardioprotective effects of dulaglutide (please see lines 421-426, 434-439), and we also addressed the effects on oxidative stress (468-470).

  1. Introduce a limitations section discussing potential biases, the limitations of animal models, and the implications for human treatment.

Limitations has been added into the discussion section. Please see lines 490-495. Limitations of this study are mainly reflected in the lack of dissection of molecular mechanisms that mediate in beneficial effect of both applied GLP-1RAs, exenatide and dulaglutide. Furthermore, it would be of interest to include more GLP-1RAs and to assess whether they achieve the cardioprotection through the same mechanisms. The lack of the translation to the human respondents also could be recognized as limitation of the study.

  1. While the animal model is suitable for this study type, translating these findings to human clinical practice requires caution; a section discussing this would be beneficial. Consider including additional studies or a meta-analysis to strengthen the context and relevance of the findings to existing literature.

Thank you for the suggestion. We addressed the translation to humans within the limitations and future directions. This could be the important direction in the future, combining the results from the basic research to the findings in human population.

  1. Improve the overall readability of the manuscript by reducing jargon and simplifying complex sentences.

Thank you for your comment. Corrected.

  1. Consider the addition of a "Future Directions" section to suggest how this research could evolve, and other related areas could be explored.

“Future directions” was added in the discussion section, after the limitations of the study. Please see lines 496-505.

  1. Ensure consistency in terminology, particularly when referring to the study groups, treatments, and outcomes.

Thank you for your comment, we checked for consistency throughout the text.

Introduction:

  1. Provide a clearer statement of the research gap your study aims to address. This will help set the context for your study's significance.

Corrected. Please see lines 92-98.

  1. Expand on the mechanism through which GLP-1 receptor agonists are hypothesized to affect ischemia-reperfusion injury, providing a stronger theoretical foundation for your research.

Several molecular mechanisms which refer to hypothesizing protective effects of GLP-1RAs are already mentioned, please see the introduction section. We further added the effects related to the inflammation, due its close relation to the oxidative stress. Please see lines 76-82.

Methods:

  1. Detail any ethical considerations or animal welfare measures taken during the study to reassure the reader of its ethical standards.

Details related to ethics can be found in a separate section at the end of the manuscript, but now we have also listed it in the material and methods section. Please see lines 522-526.

  1. Include specifics about the randomization and blinding processes to enhance the reliability and validity of your results.

Thank you for your comment. After the MetS induction, rats were randomly divided with equal distribution of animals among groups throughout the experimental period. Additionally, we employed a coding system for both pharmaceutical agents and animal enclosures, ensuring that those responsible for evaluating the study outcomes being unaware of which treatment each group receives.

  1. Clarify the selection criteria for the animal model and justify the choice of doses for exenatide and dulaglutide.

Thank you for your suggestion. Please see the material and methods section. We cited references for doses.

Discussion:

  1. Dive deeper into how your findings align or contrast with existing literature, providing a more detailed analysis of potential reasons for any discrepancies.
  2. Discuss the potential implications of your findings for future clinical applications, including any possible limitations in extrapolating animal model findings to humans.
  3. Address the potential mechanisms behind the differential effects observed between exenatide and dulaglutide, hypothesizing why one might be more effective than the other.

Thank you for your comment. Based on your suggestions above, the discussion section has been updated to incorporate a more comprehensive evaluation.

Limitations:

  1. Acknowledge limitations related to the study design, the animal model used, and the applicability of results to human populations.

Limitations has been added into the discussion section. Please see lines 490-495. Limitations of this study are mainly reflected in the lack of dissection of molecular mechanisms that mediate in beneficial effect of both applied GLP-1RAs, exenatide and dulaglutide. Furthermore, it would be of interest to include more GLP-1RAs and to assess whether they achieve the cardioprotection through the same mechanisms. The lack of the translation to the human respondents also could be recognized as limitation of the study.

  1. Discuss how these limitations could impact the interpretation of your findings and suggest areas for future research to address these gaps.

In this study we assessed the cardioprotective potential of two GLP-1RAs, exenatide and dulaglutide. The limitations of the study, which are mainly reflected in the lack of examination of molecular mechanisms, with the exception of oxidative stress, could be overcome in the following research, the aim of which would be to examine the signaling cascades that mediate the cardioprotection caused by GLP-1RAs.

Figures and Tables:

  1. Ensure all figures and tables are of high quality, clearly labelled, and referenced in the text. The black columns in figure 1 can not differentiated by the reader, please consider that in all figures.

Thank you for your comment. Figures are corrected.

General considerations on the language and grammar

Consistency in Terminology:

  • Inconsistent: "The study investigates GLP-1 receptor agonists in the first section, then examines the impacts of GLP-1RA drugs later."
  • Consistent: "The study investigates GLP-1 receptor agonists (GLP-1RAs) in the first section and continues to examine the impacts of GLP-1RAs throughout."

Corrected.

Sentence Structure:

  • Complex Sentence: "Given that metabolic syndrome increases the risk of cardiovascular disease, which is further complicated by factors such as obesity and insulin resistance, the study of GLP-1 receptor agonists becomes essential."
  • Simplified Sentences: "Metabolic syndrome increases the risk of cardiovascular disease. Factors such as obesity and insulin resistance complicate this condition. Therefore, studying GLP-1 receptor agonists is essential."

Corrected.

Grammar and Punctuation:

  • Incorrect: "After conducting the experiments, the rats were analysed, and their data was collected and reviewed."
  • Correct: "After conducting the experiments, we analyzed the rats and collected and reviewed their data."

Corrected.

Precision and Clarity:

  • Vague: "The drug had a good effect on the heart."
  • Precise: "The drug significantly reduced myocardial infarction size by 30%."

Corrected.

Style and Tone:

  • Informal: "The results were super interesting and showed we’re on the right track."
  • Formal: "The results were significant and indicated that the current research direction is promising."

Corrected.

Consistency in Formatting:

  • Inconsistent: "Table 1 shows... However, in Figure 2 we see..."
  • Consistent: "Table 1 shows... Similarly, Figure 2 illustrates..."

Corrected.

Proofreading:

  • Before: "The experiment showed that heart rate decreased in the control group."
  • After: "The experiment showed that the heart rate decreased in the control group."

Corrected.

By addressing these specific examples in your manuscript, you can enhance its overall quality and ensure it adheres to academic standards.

Comments on the Quality of English Language

The English language needs moderate revisions.

The revisions to the English language have been carefully implemented as suggested.

Please see the attachment. The answers for the Reviewer are marked in red.

Round 2

Reviewer 1 Report

Comments and Suggestions for Authors

In my view, the authors have addressed the requested modifications and recommendations, rendering this version acceptable for publication.

Author Response

Thank you for your comments.

Reviewer 2 Report

Comments and Suggestions for Authors

The authors reply to all comments, the following comments may enhance the manuscript quality:

Enlarge all figures so they can be seen easily for readers. All figure are small and could not differentiate lines 

Delve deeper into the mechanisms behind the antifibrotic effects of GLP-1 receptor agonists, exploring their impact on pathways involved in collagen synthesis and degradation, and the role of fibroblasts in cardiac remodeling.

Investigate the interaction between GLP-1 receptor agonists and other standard therapies for metabolic syndrome and cardiovascular diseases, such as statins or antihypertensive medications, to understand potential synergistic or antagonistic effects.

Consider analyzing sex-based differences in the response to GLP-1 receptor agonist treatment, as there may be variations in drug efficacy and side effects between male and female

 Assess the effects of GLP-1 receptor agonists on cardiac metabolism, including changes in glucose uptake, fatty acid oxidation, and mitochondrial function, to understand how these agents influence energy homeostasis in the heart.

Include a broader range of inflammatory biomarkers to better understand the anti-inflammatory effects of GLP-1 receptor agonists and their role in mitigating ischemia-reperfusion injury.

Explore the electrophysiological effects of GLP-1 receptor agonists on cardiac tissue, such as their impact on action potential duration, conduction velocity, and arrhythmogenic potential.

Discuss the clinical relevance of the findings in the context of patient care, including how these results could influence treatment strategies for individuals with metabolic syndrome and at risk for ischemic heart disease.

Compare the efficacy of GLP-1 receptor agonists with other cardioprotective agents, such as beta-blockers or angiotensin-converting enzyme inhibitors, to position these agents within the broader landscape of cardiovascular therapeutics.

Comments on the Quality of English Language

Minor revisions to the English language.

Author Response

Open Review

Quality of English Language

( ) I am not qualified to assess the quality of English in this paper
( ) English very difficult to understand/incomprehensible
( ) Extensive editing of English language required
( ) Moderate editing of English language required
(x) Minor editing of English language required
( ) English language fine. No issues detected

Yes

Can be improved

Must be improved

Not applicable

Does the introduction provide sufficient background and include all relevant references?

(x)

( )

( )

( )

Are all the cited references relevant to the research?

(x)

( )

( )

( )

Is the research design appropriate?

( )

(x)

( )

( )

Are the methods adequately described?

(x)

( )

( )

( )

Are the results clearly presented?

( )

(x)

( )

( )

Are the conclusions supported by the results?

(x)

( )

( )

( )

Comments and Suggestions for Authors

The authors reply to all comments, the following comments may enhance the manuscript quality:

Enlarge all figures so they can be seen easily for readers. All figure are small and could not differentiate lines.

Thank you for your suggestion. Figures are enlarged.

  1. Delve deeper into the mechanisms behind the antifibrotic effects of GLP-1 receptor agonists, exploring their impact on pathways involved in collagen synthesis and degradation, and the role of fibroblasts in cardiac remodeling.

Thank you for your suggestion. We expand the discussion, please see the lines 529-536.

  1. Investigate the interaction between GLP-1 receptor agonists and other standard therapies for metabolic syndrome and cardiovascular diseases, such as statins or antihypertensive medications, to understand potential synergistic or antagonistic effects.

Thank you for your comment. please see the lines 559-563.

  1. Consider analyzing sex-based differences in the response to GLP-1 receptor agonist treatment, as there may be variations in drug efficacy and side effects between male and female

We added gender based differences as possible model for future research. Please see the lines 554-556.

  1. Assess the effects of GLP-1 receptor agonists on cardiac metabolism, including changes in glucose uptake, fatty acid oxidation, and mitochondrial function, to understand how these agents influence energy homeostasis in the heart.

Thank you for your suggestion. We added the lines into discussion of influence of the above mentioned. Please see the lines 509-512.

  1. Include a broader range of inflammatory biomarkers to better understand the anti-inflammatory effects of GLP-1 receptor agonists and their role in mitigating ischemia-reperfusion injury.

Please see the lines 464-469.

  1. Explore the electrophysiological effects of GLP-1 receptor agonists on cardiac tissue, such as their impact on action potential duration, conduction velocity, and arrhythmogenic potential.

Please see the line into discussion section

Please see the lines 457-460.

  1. Discuss the clinical relevance of the findings in the context of patient care, including how these results could influence treatment strategies for individuals with metabolic syndrome and at risk for ischemic heart disease.

We already mentioned the translation to humans within the limitations and future directions.

  1. Compare the efficacy of GLP-1 receptor agonists with other cardioprotective agents, such as beta-blockers or angiotensin-converting enzyme inhibitors, to position these agents within the broader landscape of cardiovascular therapeutics.

Thank you for your comment. This was not the aim of our study. This potentially can be future perspective for our research. Please see section Discussion, paragraph Future research.

Comments on the Quality of English Language

Minor revisions to the English language.

Corrected.

The changes in the manuscript for the second revision were marked in yellow.
